# Therapeutic Drug Monitoring of Amphotericin-B in Plasma and Peritoneal Fluid of Pediatric Patients after Liver Transplantation: A Case Series

**DOI:** 10.3390/antibiotics11050640

**Published:** 2022-05-11

**Authors:** Francesca Tortora, Luigi Dei Giudici, Raffaele Simeoli, Fabrizio Chiusolo, Sara Cairoli, Paola Bernaschi, Roberto Bianchi, Sergio Giuseppe Picardo, Carlo Dionisi Vici, Bianca Maria Goffredo

**Affiliations:** 1Anesthesia and Critical Care Medicine, Bambino Gesù Children’s Hospital, IRCCS, 00165 Rome, Italy; francesca.tortora@opbg.net (F.T.); luigi.deigiudici@opbg.net (L.D.G.); fabrizio.chiusolo@opbg.net (F.C.); roberto.bianchi@opbg.net (R.B.); sgiuseppe.picardo@opbg.net (S.G.P.); 2Department of Pediatric Specialties and Liver-kidney Transplantation, Division of Metabolic Diseases and Drug Biology, Bambino Gesù Children’s Hospital, IRCCS, 00146 Rome, Italy; sara.cairoli@opbg.net (S.C.); carlo.dionisivici@opbg.net (C.D.V.); biancamaria.goffredo@opbg.net (B.M.G.); 3Unit of Microbiology and Diagnostic Immunology, Department of Diagnostic and Laboratory Medicine, Bambino Gesù Children’s Hospital, IRCCS, 00165 Rome, Italy; paola.bernaschi@opbg.net

**Keywords:** amphotericin-B, plasma, peritoneum, therapeutic drug monitoring, TDM, LC-MS/MS, antifungal prophylaxis, liver transplantation, pediatric

## Abstract

Fungal infections represent a serious complication during the post-liver transplantation period. Abdominal infections can occur following pre-existing colonization, surgical procedures, and permanence of abdominal tubes. In our center, liposomal amphotericin-B is used as antifungal prophylaxis in pediatric patients undergoing liver transplantation. The aim of this study is to evaluate peritoneal levels of amphotericin-B following intravenous administration. Six liver recipients received liposomal amphotericin-B. Three of them were treated as prophylaxis; meanwhile, three patients received liposomal amphotericin-B to treat *Candida albicans* infection. Plasma and peritoneal amphotericin-B levels were measured by LC-MS/MS in two consecutive samplings. C*min* (pre-dose) and C*max* (2 h after the end of infusion) were evaluated as drug exposure parameters for both plasma and peritoneum. Our results showed that peritoneal amphotericin-B levels were significantly lower than plasma and that the correlation coefficient was 0.72 (*p* = 0.03) between plasma and peritoneal C*min*. Moreover, although peritoneal levels were within the therapeutic range, they never reached the PK/PD target (Cmax/MIC > 4.5). In conclusion, PK exposure parameters could be differently used to analyze amphotericin-B concentrations in plasma and peritoneum. However, liposomal amphotericin-B should be preferred in these patients as prophylactic rather than therapeutic treatment for fungal infections.

## 1. Introduction

In recent decades, the improvements in surgical techniques and the targeted immunosuppression in pediatric liver transplantation (LT) have led to a better outcome in terms of recipient and graft survival [1]. Infections represent a significant complication that must be accounted for during the early post-surgery period and that could impact both the morbidity and mortality rate. The clinical condition of recipients (pre- and post- transplant) influences the risk of post-transplant infectious complication. In particular, cirrhotic patients demonstrate a greater susceptibility to post-transplant infections due to the persistent inflammatory state and immunological dysfunction [2]. During this time, we can identify two groups of infections in the recipients: acquired and reactivation [3]. Early infections can occur in the abdomen, blood stream, and lungs. Particularly, abdominal infections after LT can occur following pre-existing colonization/infections, surgical procedures, and permanence of abdominal tubes. Antibiotic prophylaxis, surveillance cultures, asepsis, and procedures of the patient’s isolation can reduce the incidence of post-transplantation infections.

A successful antimicrobial therapy can benefit from both antimicrobial susceptibility tests and therapeutic drug monitoring (TDM). Although TDM is not mandatory for all antimicrobials, it can be useful to improve the efficacy of antibiotics and antifungal agents while reducing at the same time the development of drug-related toxicity [4]. In particular, the effectiveness of antifungal treatments can be assessed by evaluating specific pharmacokinetic/pharmacodynamic (PK/PD) targets that change according to the different agent. For echinocandins, the ideal PK/PD target is the maximal plasma concentration (Cmax)/MIC; meanwhile, for azoles the ratio of area under the curve (AUC) on MIC should be considered as an ideal PK/PD target [5]. 

Whereas antibiotic prophylaxis is routinely prescribed according to specific protocols in the peri-operative time, antifungal prophylaxis is usually evaluated case by case.

For the adult population, the American Society of Transplantation and the Infectious Disease Society of America antifungal prophylaxis guidelines stratify liver recipients into three categories: (1) no prophylaxis; (2) prophylaxis targeted against Candida in patients with surgical complication or peri-operative Candida colonization; and (3) prophylaxis against Candida and Aspergillus in re-transplantation, renal replacement therapy, and re-operation [6,7]. No clear indications about antifungal agents, dose, or duration of therapy are specified [6]. 

Candidiasis is the most common fungal infection after LT [8]. The gastrointestinal tract is often colonized with several species of Candida and patients with end stage liver disease are more susceptible to a super colonization [9]. In fact, no consensus guidelines are available for the antifungal prophylaxis in the pediatric population undergoing LT. 

The choice of using a liposomal formulation for amphotericin-B is due to a limited risk of hepatic toxicity and a lower risk of drug–drug interactions (DDIs) with concomitant immunosuppressive therapy (i.e., tacrolimus) [10]. However, to date there are still limited data on the efficacy and safety of using liposomal amphotericin-B to treat abdominal fungal infections in pediatric patients subjected to LT. Moreover, it is worth noting that septic or critically ill patients are characterized by altered PK/PD properties for many drugs including antimicrobial agents. Therefore, this aspect leads us to increase our knowledge about the PK behavior of amphotericin-B in pediatric patients who underwent LT, in order to establish whether the use of this antifungal agent could be improved in this special population.

Recently, in our center the chance of monitoring liposomal amphotericin-B concentrations in biological fluids has been added to the routine TDM practice, allowing the quantification of the drug not only in plasma but also in other tissue compartments, such as peritoneum in the case of abdominal infections [11].

Previously, several studies reported on the peritoneal penetration of amphotericin-B [12,13] and other antifungal agents in adult patients [14]. However, no data are available on the penetration of amphotericin-B in the peritoneum of pediatric patients. 

The aim of this retrospective study is to describe the use of liposomal amphotericin-B in six pediatric liver recipients and the drug monitoring concentration in serum/abdominal fluid during the early post-operative period in the Pediatric Intensive Care Unit (PICU). Furthermore, this study aims to evaluate the possible organ toxicity and the level reached in the abdomen by using a TDM approach [11].

## 2. Materials and Methods

### 2.1. Study Design and Patients’ Characteristics

This study is a retrospective analysis of a single institution’s experience (from November 2020 to October 2021) in six children admitted in PICU after LT and for which liposomal amphotericin-B was prescribed. Patients’ demographic characteristics are reported in Table 1.

Therapeutic drug monitoring for amphotericin-B has been applied as routine clinical practice. Therefore, Ethical Review Board of our hospital has been informed about this TDM application but no formal protocol has been submitted. Plasma and peritoneal fluid samples were collected at least 4 days following the start of liposomal amphotericin-B administration and for a minimum of 4 consecutive days in order to measure drug levels at the steady state. 

Blood and peritoneal fluid samples were collected one hour before (*Cmin*) liposomal amphotericin-B administration and 2 h after (C*max*) the end of drug infusion. In all patients, liposomal amphotericin-B was intravenously administered in one hour at a dosage of 3 mg/kg according to our hospital protocol. Additional blood tests performed alongside amphotericin-B monitoring were: chemical–physical examination of both peritoneal fluid and serum, complete blood count (white blood cell count), C reactive protein (CRP), and procalcitonin (PCT) evaluation (Table 2). Whole blood was collected from an indwelling arterial line in EDTA tubes and centrifuged at 3500× *g* for 5 min to obtain plasma. The abdominal fluid was collected from an abdominal tube; all samples were fresh ones collected from the drain and immediately transported to the laboratory for analysis. Plasma samples used for selectivity and specificity evaluation were obtained from hospitalized patients receiving different antifungal treatments (but not liposomal amphotericin-B) and subjected to therapeutic drug monitoring as routine clinical practice. These patients are not included in this report. 

### 2.2. Determination of Amphotericin-B Levels by LC-MS/MS

Plasma and peritoneal fluid amphotericin-B levels were measured in the Laboratory of Metabolic Diseases and Drug Biology at Bambino Gesù Children’s Hospital in Rome. Liquid chromatography and mass spectrometry analysis were performed by using an UHPLC Agilent 1290 Infinity II coupled to a 6470 Mass Spectrometry system (Agilent Technologies, Santa Clara, CA 95051, USA) equipped with an ESI-JET-STREAM source operating in the positive ion (ESI+) mode. The software used for controlling this equipment and analyzing data was MassHunter Workstation (Agilent Technologies).

Calibrators, Quality Controls (QCs), and patients’ samples were analyzed using a validated LC-MS/MS kit (MassTox^®^ Antimycotic Drugs/EXTENDED) provided by Chromsystems (Chromsystems Instruments & Chemicals GmbH, 82166 Gräfelfing/Munich, Germany).

The assay calibration curve was linear for amphotericin-B and ranged from 0.092 to 4.88 µg/mL (Appendix A). Lower limit of quantification (LLOQ) defined by this kit was 0.050 µg/mL. Plasma and peritoneal liquid samples were prepared following manufacturer’s instructions. Samples with an amphotericin-B concentration above the higher calibration point were further diluted by using a Dilution Buffer (provided by kit) and re-analyzed once again. Each batch of patient analyses included both Low and High Quality Controls (QCs) at fixed concentrations of 1.21 and 3.62 µg/mL, respectively.

This commercial kit included plasma lyophilized calibrators and QCs and was further validated according to EMA guidelines for bioanalytical methods validation (http://www.ema.europa.eu/docs/en_GB/document_library/Scientific_guideline/2011/08/WC500109686.pdf accessed on 21 July 2011).

For this purpose, we have evaluated selectivity and specificity by analyzing six different plasma samples of patients who were assuming other antifungals (i.e., Isavuconazole) but not liposomal amphotericin-B. The median signal of these blank samples was below 20% of the LLOQ, thus ensuring the selectivity of the method. Representative chromatograms of blank sample + isotopically labelled internal standard (IS) and amphotericin-B+IS are reported in Appendix A, respectively. Accuracy and precision were analyzed on QCs samples. Accuracy was reported as Bias %; meanwhile, precision was defined as % coefficient of variation (CV) for both High and Low QC. In particular, Bias % was −17.35 and −10.85 for Low and High QCs, respectively. Similarly, % CV was 14.85 for Low and 8.61 for High QC.

### 2.3. Statistical Analysis

All statistical analyses and graphs were performed using Graph-Pad Prism 7.0 (Graph-Pad software Inc., San Diego, CA, USA). PK parameters were analyzed using descriptive statistics. For this study, no formal power calculation was made. Median with interquartile range (IQR) was used for not-normally distributed measurements. Penetration ratio was expressed as geometric mean with 95% confidence intervals. Mann–Whitney was used as nonparametric test to compare two groups of data. Statistical significance was set at *p* < 0.05.

## 3. Results

The six patients required antifungal therapy as prophylaxis (*n* = 3) or for treatment of peritoneal Candida infection (*n* = 3). Patients’ demographic characteristics are shown in Table 1. All patients were cirrhotic. Liposomal amphotericin-B dosage was increased from 3 mg/kg to 5 mg/kg only in Patient 1 due to persistence of Candida infection after 34 days from administration starting. Antifungal prophylaxis was prescribed in Patients 2, 4, and 6 due to biliary tract surgery and abdomen revision after LT (Patients 2 and 4) and for preoperative admission in PICU where Patient 6 received continuous kidney replacement therapy. Candida was also detected in pleural fluid in Patient 3. For Patients 1, 3, and 5, MIC values of Candida isolates for amphotericin-B were 0.125–1.0, 0.25, and 0.25 µg/mL, respectively. Standard recommendations suggest as PK/PD target for amphotericin-B a Cmax/MIC > 4.5 [11]. All laboratory results for blood and peritoneal fluid tests are shown in Table 2. Except for Patient 4, all studied patients showed high level of white cell count/polymorphonuclear white cell count in peritoneal fluid.

Measured amphotericin-B concentrations in blood and peritoneum samples are shown in Table 3. For the first TDM, performed at 10 ± 4 (mean ± SE) days after starting therapy, the median amphotericin-B C*min* in plasma and peritoneum were 1.33 (IQR, 0.52–5.36) µg/mL and 0.60 (IQR, 0.29–1.10) µg/mL, respectively. Similarly, the median amphotericin-B C*max* in plasma and peritoneum were 16.71 (IQR, 8.01–22.05) and 0.47 (IQR, 0.28–0.90) µg/mL, respectively. 

As concerns the second TDM, performed at 17 ± 7 days after starting therapy, median amphotericin-B plasma C*min* and C*max* were 0.81 (IQR, 0.15–2.06) and 6.65 (IQR, 5.88–7.01) µg/mL, respectively. For this second measurement, median amphotericin-B levels in peritoneum were 0.39 (IQR, 0.32–0.43) for C*min* and 0.37 (IQR, 0.25–0.47) for C*max*. 

Exclusively for the first assessment, peritoneal C*max* levels were significantly lower than plasma levels (*p* < 0.01) (Figure 1A). However, both peritoneum C*min* and C*max* were in the therapeutic range (0.2–3.0 µg/mL) established for our TDM laboratory [15,16]. Similarly, for the second measurement, amphotericin-B peritoneal levels were within the therapeutic range for both C*min* and C*max*.
Figure 1Evaluation of amphotericin-B levels in both plasma and peritoneum. (**A**) Comparison of amphotericin-B C*max* in plasma and peritoneal fluid. Data are expressed as median with interquartile range, ** *p* < 0.01; (**B**) Linear regression model with the best fit line (black) and 95% CI (dotted blue lines), describing correlation between amphotericin-B C*min* in plasma and peritoneum; (**C**) Linear regression model with the best fit line (black) and 95% CI (dotted blue lines), describing correlation between amphotericin-B C*max* in plasma and peritoneum. The asterisk (*) symbol indicates “times” the values on the *X* axis.
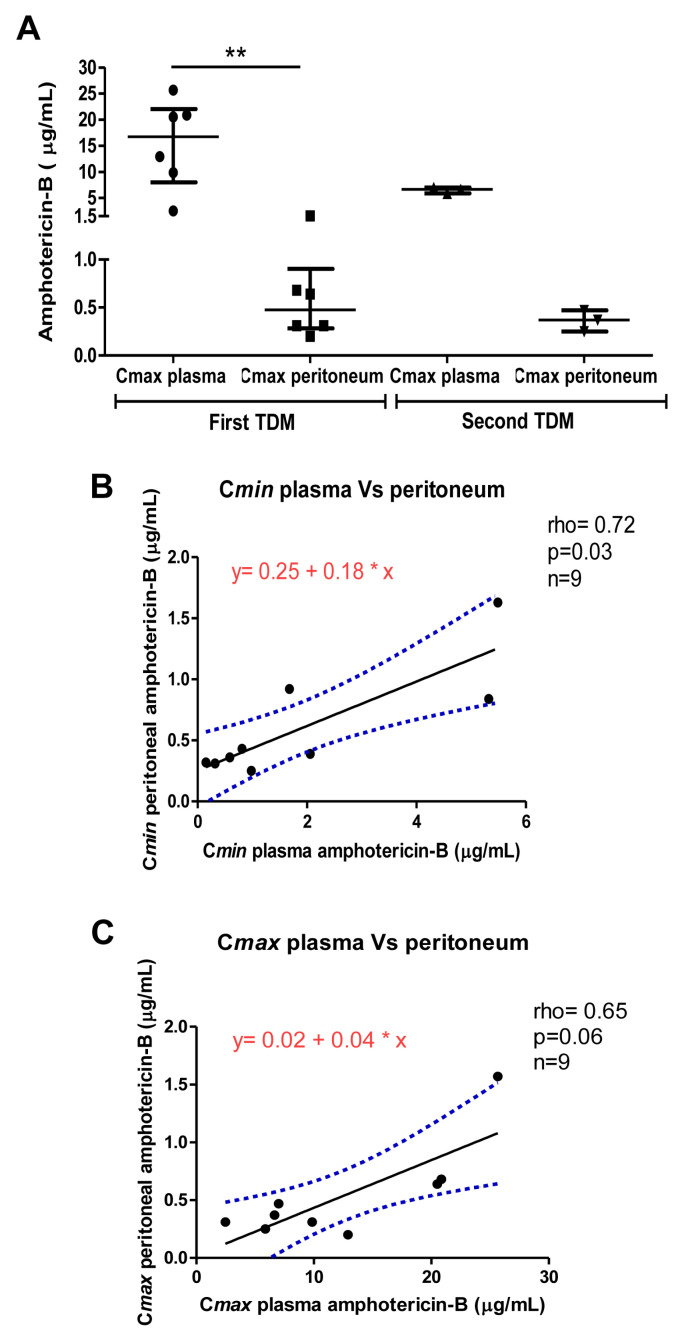


Evaluation of penetration ratio (peritoneal/plasma levels) from both first and second measurements was significantly higher (*p* < 0.001) for C*min* (geometric mean; 95% CI= 0.46; 0.24–0.86) compared to C*max* (geometric mean; 95% CI= 0.04; 0.03–0.07).

A significant correlation was found between plasma and peritoneal C*min* from both first and second TDM (Spearman r = 0.72, *p* = 0.03) (Figure 1B). Conversely, plasma and peritoneal C*max* were not significantly correlated (Spearman r = 0.65, *p* = 0.06) (Figure 1C). 

Endovenous administration of liposomal amphotericin-B showed a dose proportionality when considering C*min* as an exposure parameter for plasma (Spearman r = 0.97, *p* < 0.01) (Figure 2). 

## 4. Discussion

In this study, we aimed to assess the relation between plasma and peritoneal levels of amphotericin-B in pediatric patients. We show that amphotericin-B peritoneal levels are significantly lower than plasma levels. In particular, this difference appears when C*max* for both plasma and peritoneum are compared. However, penetration ratio (peritoneal/plasma levels) is significantly higher when evaluating C*min* rather than C*max*. The reasons for this low penetration of amphotericin-B in peritoneum have been already debated in the literature [17]. PK properties represent an important aspect that limits amphotericin-B penetration. Because it has poor water solubility, a distribution volume (Vd) is limited to the plasmatic fraction in which liposomal amphotericin-B shows a high protein bound percentage (91–95%) and forms high molecular weight complexes that lead to a half-life of 15 days [13].

Another aspect that could affect amphotericin-B levels in peritoneum is the severity of peritonitis because someone can postulate that an inflammatory condition of the peritoneal membrane could allow an easy penetration of drugs from blood to the peritoneal compartment [11]. However, we postulate the opposite because CRP levels are inversely correlated to both amphotericin-B C*min* (Spearman r = −0.65, *p* = 0.17) and C*max* (Spearman r= −0.55, *p* = 0.30) in the peritoneal fluid (data not shown). On the other hand, the diagnosis of peritonitis is generally made in patients with polymorphonuclear white cells (PMNWC) count > 250/mm^3^ in peritoneal fluid and in the presence of fever, leukocytosis, and increases in CRP and PCT [18]. Patients described in this study developed fever; however, only two required infusion of vasopressor drugs. Therefore, we believe that fever is a non-specific symptom that is not exclusively related to conditions such as infection and sepsis. Similarly, leukocytosis and high levels of CRP and PCT may be ascribed to multiple factors [19]. 

Candida albicans was isolated from abdominal fluid samples of three patients (Patients 1, 3, and 5) whereas bacteria were found on examination of the peritoneal fluid in five patients. In particular, bacterial species found were Pseudomonas aeruginosa, Klebsiella pneumoniae, Enterobacter cloacae, Enterococcus faecium, and faecalis (Table 1). All patients received a targeted antibiotic therapy and, for two patients, the antibiotic concentrations in peritoneal fluids were similar to the blood levels. Only one patient developed positive blood cultures for Candida albicans during hospitalization (Patient 1). No patients died from sepsis.

The MIC for amphotericin-B has been reported in the range of 0.125–1.0 µg/mL [15,16]; meanwhile, standard recommendations suggest as the PK/PD target for amphotericin-B a Cmax/MIC > 4.5 [11]. Based on our results, although amphotericin-B levels in the peritoneal fluid were within the target therapeutic range (0.2–3.0 µg/mL), they never reached the PK/PD index at the site of action. This could perhaps be explained by a suboptimal dosage of liposomal amphotericin-B. However, considering that two patients out of three with confirmed Candida infection showed an improvement of clinical conditions even without a dosage increase, it is reasonable to think that in absence of consensus data and guidelines about antifungal prophylaxis in pediatric patients during the post-LT period, a further increase in the liposomal amphotericin-B dose could expose patients to an unjustified risk of renal toxicity. Additionally, it is worth noting that the importance of intraperitoneal drug levels could be debated. In fact, on one hand the antifungal should be opportunely concentrated at the site of infection, but on the other hand a clear relationship between peritoneal amphotericin-B and the clinical outcome has not yet been explored.

Our results are consistent with previously published reports in which amphotericin-B peritoneal levels were significantly lower or undetectable compared to those measured in plasma [12,13]. Similarly, van der Voort et al. [17] have previously shown that continuous infusion of amphotericin-B in adult critically ill patients affected by Candida peritonitis produced peritoneal drug levels lower than serum. In this study, we report similar results through administering liposomal amphotericin-B as one-hour iv infusion. However, this is the first report in which peritoneal penetration of liposomal amphotericin-B has been studied in a special population, such as pediatric liver recipients subjected to a prophylactic or therapeutic antifungal treatment. Obviously, we cannot exclude that our study has some limitations.

In fact, cases described in this study are very heterogeneous in age, weight, and underlying disease. Additionally, each patient received different pharmacological and antimicrobial treatments alongside liposomal amphotericin-B administration.

Another issue is that amphotericin-B measurement could be affected by the presence of liposomal capsules; therefore, determination of the drug fraction that has been liberated from its lipid encapsulation and that can be measured in the plasmatic and peritoneal compartments is not always straightforward. However, the bioanalytical method used in this study to measure amphotericin-B levels in both plasma and peritoneum allows for this issue to be overcome. This is because during the sample preparation phase, liposomal fractions were disrupted, which led to an easier determination of liberated drug levels.

In conclusion, our study shows that amphotericin-B levels are significantly lower in the peritoneal fluid compared to plasma levels, although a positive correlation is established between plasma and peritoneal C*min*. However, even if peritoneal concentrations of amphotericin-B were within the therapeutic range established for our TDM laboratory, they were not sufficient to reach the PK/PD target for amphotericin-B (C*max*/MIC > 4.5). On this point, it is worth noting that to allow an adequate distribution of amphotericin-B into peritoneum, a sampling time later than 2 h post-infusion could be considered for evaluation of C*max* in peritonal fluid. 

In our pediatric population, PK exposure parameters could be differently used to analyze amphotericin-B levels in both plasma and peritoneum. From our results, C*min* seems to be a good candidate to calculate the penetration ratio and to correlate plasma vs. peritoneal drug levels. 

Similarly, C*max* could be the ideal target to compare amphotericin-B levels in both plasma and peritoneal fluid. On the other hand, C*min* could be used to analyze dose vs. plasma levels proportionally.

Our experience confirms that liposomal amphotericin is an option for antifungal prophylaxis during the post-operative period of liver transplantation. The careful infectious monitoring with culture tests should also be extended to the peritoneal fluid (when abdominal tubes are placed on it) so as to prescribe an antifungal agent able to reach the therapeutic PK/PD target and C*min* in the abdomen, when peritoneal fluid culture is positive to fungi.

Although at present there are no general recommendations for routine TDM of amphotericin B, based on this report we believe that in the absence of clear guidelines, monitoring of amphotericin-B concentrations may represent an improvement in clinical care of special subjects, such as critically ill pediatric patients. 

Finally, we are aware that this is an observational report and that proper clinical studies are necessary to evaluate the PK properties of amphotericin-B in both plasma and peritoneal fluid, following intravenous administration of prophylactic or therapeutic treatment of fungal infections in pediatric patients who have undergone LT. However, in the absence of specific consensus guidelines and proper PK information, this study could be preparatory for future investigations on liposomal amphotericin-B use in pediatric patients.

## Figures and Tables

**Figure 2 antibiotics-11-00640-f002:**
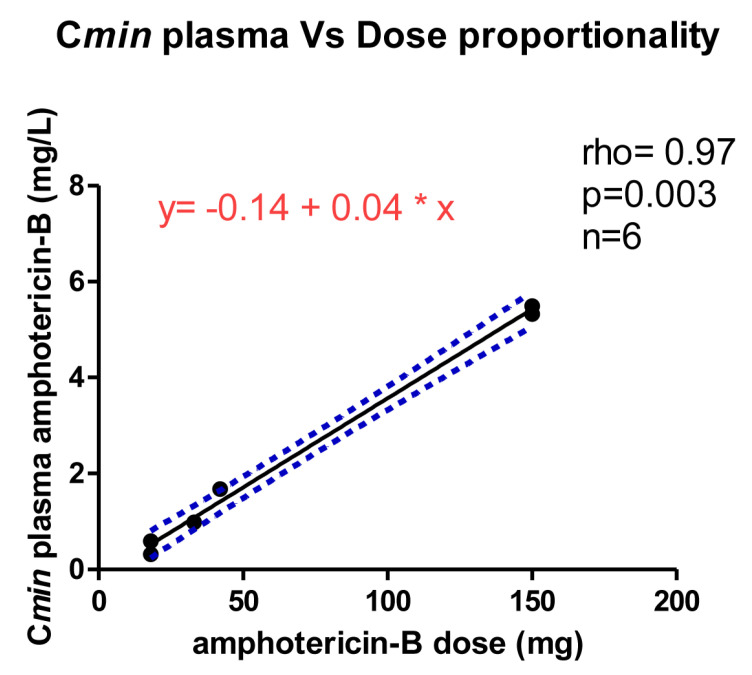
Dose proportionality evaluation. Linear regression model with the best fit line (black) and 95% CI (dotted blue lines), describing correlation between amphotericin-B C*min* in plasma and dose administered. The asterisk (*) indicates “times” the value on the *X* axis.

**Table 1 antibiotics-11-00640-t001:** Patient demographic characteristics.

Patients	Age ^1^	Weight ^2^	Primary Disease	Cirrhosis	Candida Albicans in Peritoneum	Candida Albicans in Blood	Bacteria Peritoneum	Antimicrobial Therapy
Patient 1	242	50	Biliary cirrhosis	yes	yes	yes	Enterococcus Faecalis Enterobacter Cloacae Klebsiella Pneumonie	vancomycin meropenem tigecycline
Patient 2	5	6	Biliary atresia	yes	no	no	Enterococcus Faecalis Klebsiella Pneumoniae	teicoplanin meropenem amikacin
Patient 3	18	11	Biliary atresia	yes	yes	no	Enterococcus Faecium Pseudomonas Aeruginosa	vancomycin meropenem
Patient 4	5	6	Biliary atresia	yes	no	no	Enterococcus Faecalis	meropenem teicoplanin levofloxacin
Patient 5	92	14	Alagille syndrome	yes	yes	no	Enterococcus Faecium Pseudomonas Aeruginosa	vancomycin meropenem amikacin
Patient 6	236	50	Biliary cirrhosis	yes	no	no	Enterococcus Faecium	teicoplanin

^1^ Age expressed in months; ^2^ Weight expressed in kilograms.

**Table 2 antibiotics-11-00640-t002:** Blood tests associated with dosage of amphotericin-B levels.

Patient	WC Blood cell/µL	WC/PMN Peritoneum mm^3^	CRP mg/dL	PCT ng/mL
Patient 1 1st TDM	31,190	15/7	8.8	2.94
Patient 1 2nd TDM	19,890	6867/4666	12.59	2.97
Patient 2 1st TDM	8690	1793/987	5.44	0.29
Patient 3 1st TDM	10,050	996/628	9.88	2.72
Patient 3 2nd TDM	6650	764/542	4.33	0.14
Patient 4 1st TDM	8780	82/36	5.26	0.51
Patient 4 2nd TDM	11,370	23/16	6.28	0.91
Patient 5 1st TDM	6480	450/387	3.89	0.18
Patient 6 1st TDM	13,111	809/679	0.57	1.06
Patient 6 2nd TDM	26,840	671/553	4.6	0.6

CRP: C reactive protein; PCT: procalcitonin; TDM: Therapeutic Drug Monitoring; WC/PMN: white cell count/polymorphonuclear white cell count.

**Table 3 antibiotics-11-00640-t003:** Levels of amphotericin-B in the blood (Plasma) and peritoneal (Peritoneum) fluid.

	First TDM						Second TDM					
Patient	Dosage mg/kg	Days after Amphotericin Start	Cmin Plasma	Cmin Peritoneum	Cmax Plasma	Cmax Peritoneum	Dosage mg/kg	Days after Amphotericin Start	Cmin Plasma	Cmin Peritoneum	Cmax Plasma	Cmax Peritoneum
1	3	34	5.32	0.84	20.51	0.64	5	41	4.55	0.13	--º	0.21
2	3	9	0.59	0.36	2.47	0.31	3	--º	--º	--º	--º	--º
3	3	6	0.98	0.25	9.86	0.31	3	10	0.81	0.43	7.01	0.47
4	3	6	0.32	0.31	12.91	0.2	3	10	0.15	0.32	5.88	0.25
5	3	4	1.68	0.92	20.85	0.68	3	--º	--º	--º	--º	--º
6	3	5	5.49	1.63	25.66	1.57	3	10	2.06	0.39	6.65	0.37

TDM: Therapeutic Drug Monitoring; º data not available.

## Data Availability

Data are available on reasonable request from the corresponding author.

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
