# Peer review of "Therapeutic Drug Monitoring of Amphotericin-B in Plasma and Peritoneal Fluid of Pediatric Patients after Liver Transplantation: A Case Series"

_antibiotics, 2022, doi:10.3390/antibiotics11050640_

Round 1

Reviewer 1 Report

The manuscript by Tortora et al is well written and mostly comprehensible.

Bacteria and yeasts are written with their genus name in upper case and their species name in lower case e.g. Candida albicans. Unfortunately, the species are incorrectly listed throughout the manuscript.

Lines 66-70 are redundant

The study uses liposomal Amphotericin B, but sometimes there seems to be confusion with Amphotericin B deoxycholate. Perhaps the authors could emphasise this a little more

Line 144-145 With what liquid were the samples diluted?

Line172 prophylaxis

The MIC of the Candida isolates for amphothericin B are not mentioned, so how can the authors be sure that the Cmax/MIC ratio of >4.5 will not be achieved?

It is not clear to me what the purpose of Table 2 is and what information the reader can derive from it and whether it relates to the amphothericin B levels.

Author Response

Reviewer #1

Comments and Suggestions for Authors

The manuscript by Tortora et al is well written and mostly comprehensible.

1) Bacteria and yeasts are written with their genus name in upper case and their species name in lower case e.g. Candida albicans. Unfortunately, the species are incorrectly listed throughout the manuscript.

Thank you for your comment. Please, see lines 25, 276, 278, 279, 296.

2) Lines 66-70 are redundant

Thank you for your comment. Following your advice, we have now removed lines 66-70.

3) The study uses liposomal Amphotericin B, but sometimes there seems to be confusion with Amphotericin B deoxycholate. Perhaps the authors could emphasise this a little more.

Thank you for highlighting this. We have now underlined the use of liposomal amphoreticin-B rather than deoxycholate. Please see lines 24, 25, 97, 128, 130, 132, 198, 246, 260, 303, 307, 319, 325, 355, 366.

4) Line 144-145 With what liquid were the samples diluted?

Thank you for your insightful question. We have included more details on sample dilution in Materials and Methods section.

5) Line172 prophylaxis

Thank you. We have followed your suggestion. Please see line 76.

6) The MIC of the Candida isolates for amphothericin B are not mentioned, so how can the authors be sure that the Cmax/MIC ratio of >4.5 will not be achieved?

Thank you for this question. We have tested MIC of the Candida isolates only for patients 1, 3 and 5, because only for these patients we isolated Candida. For patients 2, 4 and 6 liposomal amphotericin-B was prescribed as prophylaxis. Please see lines 203-205.

7) It is not clear to me what the purpose of Table 2 is and what information the reader can derive from it and whether it relates to the amphothericin B levels.

Thank you for your comment. We have simplified table 2. We have removed transaminase and creatininemia. We have considered table 2 to demonstrate that all patients (except patient 4) showed high level of white cell count/polymorphonuclear white cell count in peritoneal fluid with high level of CRP to demonstrate the inflammatory condition of peritoneal membrane. Please see lines 207-208 and 263-275.

Reviewer 2 Report

The authors provide data on amphotericin B penetration into peritoneal fluid after livertransplantation in 6 children. These data are scarce and therefore potentially relevant for future therapeutic decisions.

There are, however, some aspects that need to be addressed.

General comments:

At present there are no general recommendations for routine TDM of amphotericin B. The authors state however that  the TDM of amphotericin B in plasma and peritoneal fluid  is part of their routine care in liver transplant children. Please explain and discuss the reasoning, since this seems to be the reason to abstain from informed consent and ethical vote.

One crucial methodological aspect is the sampling time of Cmax-samples. The authors chose 2 hours post infusion as Cmax sampling point for plasma and peritoneal fluid samples. To assess the plasma-tissue-penetration ratio it is important to use correspondant samples, but this is not necessary for the evaluation of the relevant pk/pd – ratios: Therefore the authors should provide data or reference, that the Cmax in plasma is reached at 2 hours after infusion and the authors should provide data or reference, that the Cmax in peritoneal fluid is reached at 2 hours after infusion. If no data is available, the fact that the sampling point for Cmax was chosen arbitrarily should be discussed. I anticipate that the Cmax in plasma is some time earlier than in peritoneal fluid due to the complex distribution.

The correct sampling point for C max is important, since the conclusions of the authors are based on the fact, that Cmax/MIC ratio in peritoneal fluid is not reaching the target.

Specific comments:

page 3 line 123 (and throughout the text): “….was administered as a bolus in one hour….”- this is not a bolus, use the term infusion or intravenous drip

page 3 and 4 Determination of amphotericin B levels by LC-MS/MS: The authors report that they have used the Chromsystems MassTox Antimycotic Drugs/EXTENDED Assay. According to the manufacturer this assay delivers only qualitative (no quantification) information regarding amphotericin B. The authors therefore must have validated a quantitative approach using the commercial qualitative assay. The authors should provide some more data on the validation (e.g. sample preparation, LLOQ, specificity, selectivity). The validation data they provided was derived from plasma samples? Please report validation of the assay in peritoneal fluid.

Page 6 lines 204-209 and figures B, C and D: The authors provide data, that Cmax of amphotericin B in plasma and peritoneal fluid are not correlated. During the early phase after amphotericin infusion the penetration ratio is low with relatively high plasma concentration and low peritoneal concentration (figure D). In my opinion this information could be anticipated since distribution at plasma Cmax is probably not completed. The rather better data at Cmin suggests that the pharmacokinetic profile of distribution into peritoneal fluid differs from the pharmacokinetic profile in plasma. Therefore, I would recommend to provide the penetration ratio in text and omit the figures.

Page 8 lines 285 – 290: the authors suggest in their discussion that they measured free amphotericin B concentrations – how was that achieved? Please provide data in the methods section – did the authors use ultrafiltration of the samples?

Author Response

Reviewer #2

Comments and Suggestions for Authors

 General comments:

1) At present there are no general recommendations for routine TDM of amphotericin B. The authors state however that the TDM of amphotericin B in plasma and peritoneal fluid is part of their routine care in liver transplant children. Please explain and discuss the reasoning, since this seems to be the reason to abstain from informed consent and ethical vote.

Thank you for your comment. For antibiotics, measurement of plasma concentrations is performed in order to guarantee that a specific PK/PD target has been reached and to tailor an appropriate dosing regimen limiting adverse reactions.

In special conditions, such as pediatric liver transplant with peritoneal infective/inflammatory state (during pediatric intensive care stay), we believe that TDM could help clinicians to assess whether adequate drug concentrations have been reached at the site of infection, and the attainment of PK/PD targets.  

Alternatively, TDM could suggest a switch to another antifungal agent. Actually, in absence of clear recommendations about TDM of amphotericin-B, we believe that evaluation of drug exposure-response relationships may represent an improvement in clinical care of special subjects such as critically ill pediatric patients.

Ref.: Ashbee HR, et al. Therapeutic drug monitoring (TDM) of antifungal agents: guidelines from the British Society for Medical Mycology. J Antimicrob Chemother. 2014. PMID: 24379304 

2) One crucial methodological aspect is the sampling time of Cmax-samples. The authors chose 2 hours post infusion as Cmax sampling point for plasma and peritoneal fluid samples. To assess the plasma-tissue-penetration ratio it is important to use correspondant samples, but this is not necessary for the evaluation of the relevant pk/pd – ratios: Therefore the authors should provide data or reference, that the Cmax in plasma is reached at 2 hours after infusion and the authors should provide data or reference, that the Cmax in peritoneal fluid is reached at 2 hours after infusion. If no data is available, the fact that the sampling point for Cmax was chosen arbitrarily should be discussed. I anticipate that the Cmax in plasma is some time earlier than in peritoneal fluid due to the complex distribution.

The correct sampling point for Cmax is important, since the conclusions of the authors are based on the fact, that Cmax/MIC ratio in peritoneal fluid is not reaching the target.

We’d like to thank the Reviewer for raising this point. In order to evaluate amphotericin-B Cmax we have collected blood and peritoneal fluid 2 hours after the end of one hour AmBisome infusion since a Tmax (h) of 3.3 ± 0.96 has been previously reported for liposomal amphotericin- B (Bekersky I, Fielding RM, Dressler DE, Lee JW, Buell DN, Walsh TJ. Plasma protein binding of amphotericin B and pharmacokinetics of bound versus unbound amphotericin B after administration of intravenous liposomal amphotericin B (AmBisome) and amphotericin B deoxycholate. Antimicrob Agents Chemother. 2002 Mar;46(3):834-40).

 However, this reference study has been conducted on adults patients receiving liposomal amphotericin B as a single 2-h infusion (2 mg/Kg) whilst in our report pediatric patients have been given liposomal amphotericin-B as 1-hour infusion at 3 mg/Kg according to our hospital protocol. Therefore, in absence of proper PK studies on Amphotericin-B use in pediatric patients, we have used a slightly earlier time point for Cmax in both plasma and peritoneal fluid.     

We are in agreement with Reviewer that a later sampling point should be considered at least  for peritoneum in order to allow a better distribution of amphotericin B in this compartment ; however, the aim of this report was not to study PK/PD behaviour of amphotericin-B in peritoneal fluid but to correlate plasma and intraperitoneal levels during intravenous treatment of pediatric patients, and to calculate the penetration ratio as previously published (van der Voort PH, Boerma EC, Yska JP. Serum and intraperitoneal levels of amphotericin B and flucytosine during intravenous treatment of critically ill patients with Candida peritonitis. J Antimicrob Chemother. 2007 May;59(5):952-6). However, we have now included a sentence in the Discussion to clarify this point.    

Specific comments:

  • page 3 line 123 (and throughout the text): “….was administered as a bolus in one hour….”- this is not a bolus, use the term infusion or intravenous drip

Thank you for your comment. Please see lines 131 and 318.

  • page 3 and 4 Determination of amphotericin B levels by LC-MS/MS: The authors report that they have used the Chromsystems MassTox Antimycotic Drugs/EXTENDED Assay. According to the manufacturer this assay delivers only qualitative (no quantification) information regarding amphotericin B. The authors therefore must have validated a quantitative approach using the commercial qualitative assay. The authors should provide some more data on the validation (e.g. sample preparation, LLOQ, specificity, selectivity). The validation data they provided was derived from plasma samples? Please report validation of the assay in peritoneal fluid.

We thank the Reviewer for raising this point. Following your suggestion we have included in the Methods section selectivity and specificity data, which were evaluated in accordance to European Medicines Agency. 2011. Guideline on bioanalytical method validation. European Medicines Agency, London, United Kingdom:http://www.ema.europa.eu/docs/en_GB/document_library/Scientific_guideline/2011/08/WC500109686.pdf.

A Supplementary Figure including the calibration curve for amphotericin-B, chromatograms of a blank sample+ the isotopically labelled internal standard (IS) and an amphotericin-B sample + IS, has been now added to the main manuscript.

Method validation has been performed on plasma matrix. Due to the ethical concerns in obtaining blank peritoneal fluids, it was not possible to quantitatively validate the amphotericin-B bioanalytical method on peritoneal liquid. This is in agreement with previous reports in which amphotericin- B levels have been measured in pleural effusions and peritoneal liquids by using analytical methods validated on serum or plasma.

Ref:

van der Voort PH, Boerma EC, Yska JP. Serum and intraperitoneal levels of amphotericin B and flucytosine during intravenous treatment of critically ill patients with Candida peritonitis. J Antimicrob Chemother. 2007 May;59(5):952-6.

Weiler S, Bellmann-Weiler R, Joannidis M, Bellmann R. Penetration of amphotericin B lipid formulations into pleural effusion. Antimicrob Agents Chemother. 2007 Nov;51(11):4211-3.

  • Page 6 lines 204-209 and figures B, C and D: The authors provide data, that Cmax of amphotericin B in plasma and peritoneal fluid are not correlated. During the early phase after amphotericin infusion the penetration ratio is low with relatively high plasma concentration and low peritoneal concentration (figure D). In my opinion this information could be anticipated since distribution at plasma Cmax is probably not completed. The rather better data at Cmin suggests that the pharmacokinetic profile of distribution into peritoneal fluid differs from the pharmacokinetic profile in plasma. Therefore, I would recommend to provide the penetration ratio in text and omit the figures.

Thanks for this suggestion. Following Reviewer’s recommendation, penetration ratio has been removed from Figure 1 and anticipated in the text.

  • Page 8 lines 285 – 290: the authors suggest in their discussion that they measured free amphotericin B concentrations – how was that achieved? Please provide data in the methods section – did the authors use ultrafiltration of the samples?

We do apologies for this misleading point. The term “free” is not intended as amphotericin-B unbound fraction. In fact, we have not performed any ultrafiltration of plasma or peritoneal fluid samples in order to measure unbound or protein-bound amphotericin-B fractions. Based on our intent, the term “free” should indicate the amount of amphotericin-B liberated from its lipid encapsulation. However, this point has been better clarified in the Discussion section and the word “free” has been now replaced.   

Round 2

Reviewer 2 Report

The authors responded adequately to the reviews comments.